# Social context of contraceptive use transition among sexually active women in Zambia (1992–2018): A decomposition analysis

**Million Phiri**[1,2]*, **Clifford Odimegwu**[1], **Yemi Adewoyin**[1,3]

**1** Demography and Population Studies Programme, Schools of Public Health and Social Sciences, University of the Witwatersrand, Johannesburg, South Africa, **2** Department of Demography, Population Sciences, Monitoring and Evaluation, School of Humanities and Social Sciences, University of Zambia, Lusaka, Zambia, **3** Department of Geography, University of Nigeria, Nsukka, Nigeria

\* millsphiri@gmail.com

**Data Availability Statement:** Data used in our study is publicly available at IPUMS DHS website (https://www.idhsdata.org/idhs/) or DHS program website (https://dhsprogram.com/).

## Abstract

### Background

Contraception is an important public health initiative for addressing maternal health outcomes associated with unplanned pregnancies, unsafe abortions and maternal deaths. Although contraceptive use has been on the rise globally, the observed increases in sub-Saharan Africa (SSA) are sub-optimal and vary among countries. In Zambia, drivers of contraceptive use transition are not well documented. Thus, this study examined the drivers of contraceptive use change among sexually active women in Zambia between 1992 and 2018.

### Methods

Data came from the six Zambia Demographic and Health Surveys conducted between 1992 and 2018. A sample of 44,762 fecund sexually active women aged 15–49 years was analysed using multivariable Blinder Oaxaca decomposition regression analysis. Analysis took into account the complex survey design. Results were presented using adjusted coefficients, their 95% confidence intervals, and percentages.

### Results

The prevalence of contraceptive use among sexually active women increased significantly by 30.8 percentage points from 14.2% (95% CI: 12.8, 15.6) to 45.0% (95% CI: 43.6, 46.4) during the period 1992 to 2018. The major share of the increase happened during the period 1992–1996 (10.2%) while the least increase occurred between 2013 and 2018 (0.2%). Overall, about 15% of the increase in the prevalence of contraceptive use was attributable to changes in the compositional characteristics of women. On the other hand, 85% of the increase was due to change in contraceptive behaviour of sexually active women. Changes in women's compositional characteristics such as secondary education (5.84%), fertility preference (5.63%), number of living children (3.30%) and experience of child mortality (7.68%) were associated with the increase in contraceptive use prevalence.

**Funding:** The authors received no specific funding for this work.

**Competing interests:** The authors have declared that no competing interests exist.

**Abbreviations:** CI, Confidence Interval; EA, Enumeration Area; FP, Family Planning; SDG, Sustainable Development Goal; SRH, Sexual and Reproductive Health; SSA, Sub-Saharan Africa; UNFPA, United Nations Population Fund; UNICEF, United Nations Children Fund; USAID, United States Aid for International Development; WHO, World Health Organisation; ZDHS, Zambia Demographic and Health Survey.

## Conclusion

Change in contraceptive behaviour of sexually active women contributed largely to the observed increase in contraceptive use prevalence in Zambia. Increase in the proportion of women attaining secondary education, decrease in the percentage of women who want large families and improvement in child survival were the major compositional factors driving the rise in contraceptive use. The findings imply that increasing investment in education sector and enhancing existing family planning programmes has the potential to further improve contraceptive use prevalence in Zambia.

## Introduction

Contraception is a very important global public health measure to improve the health of mothers and prevent unplanned births, unsafe abortions, and maternal deaths [1–4]. Contraceptive use has increased in many regions of the world, particularly in Asia, Latin America, and higher-income countries. However, sub-Saharan Africa (SSA) continues to have a low rate of contraceptive use [5–8]. This indicates regional disparities in access to and utilisation of contraceptive methods. Although contraceptive use has been rising generally in SSA, the observed increases are sub-optimal and vary among countries [9–11]. Literature show that contraceptive use transition differs across countries in SSA. While there has been an increase in contraceptive use in some countries, in others there has been a stall and yet a decrease is noted in some countries [12].

While contraceptive use has increased in many SSA countries, there are still significant proportions of women with unmet need, particularly in certain regions of Africa [13, 14]. Therefore, efforts to reduce unmet need and increase contraceptive use should focus on reaching marginalised populations, such as adolescents and young women, individuals who are less educated and those who reside in remote areas. Addressing the unmet need for family planning (FP) is one key priority area for the Family Planning 2030 agenda (FP2030) [15]. Many governments in SSA are investing local resources to strengthen FP programmes. Strong political commitment, public education campaigns, improved access to SRH services, and increased availability of contraception methods have contributed to success of family planning initiatives in various countries [12, 16–19].

Fertility rate in Zambia has remained resilient for four decades now, reducing from 6.5 to 4.7 between 1992 and 2018 [20, 21]. The negative consequences of the current fertility landscape in Zambia is evident in the high rates of maternal and infant mortality and unsafe abortions [22, 23]. Modern contraceptive prevalence rate has been increasing steadily among married women from 9% in 1992 to 25% in 2001. It further increased from 33% in 2007 to 48% in 2018 [20]. The government of Zambia with assistance from the United Nations Population Fund (UNFPA) and other stakeholders has invested huge resources in the country FP programming [24, 25]. In 2012, the government launched a multi-sectoral integrated FP scale-up plan following the commitment to implement the resolutions of the London 2012 FP summit [23]. The implemented FP strategies included efforts to strengthen demand generation and behaviour change communication, through provision of sexual and reproductive health education and services to teenagers and young people, and improved the distribution and availability of FP commodities in the country [23, 26, 27]. Further, these strategies are aimed at enabling women to make informed choices regarding their own health.

Several studies on contraception and family planning conducted in SSA, and specifically in Zambia have suggested that demographic, socio-demographic, economic and contextual factors are associated with utilisation of contraceptive methods among reproductive-aged women [28–34]. However, there is a paucity of knowledge to adequately addresses factors that explain observed transition in contraceptive use in Zambia. The influence of both individual level and contextual factors such as increased education level, employment opportunities, changes in fertility experience and preference and how they drive contraceptive use dynamics remain to be explored in Zambia. This is because prior studies on contraception have largely focused on examining factors associated with use than contraceptive use transition. Contraceptive use transition among women could result from an improvement in social context such as improvement in education, women's empowerment and contraceptive behaviour change. Thus, it remained unknown whether the observed contraceptive use in Zambia is due to changing social context or reproductive behaviour change. This information is required to inform appropriate family planning policy re-design for improved service delivery. Therefore, this study sought to explore how individual and contextual factors have influenced change in utilisation of contraceptive methods of sexually active women in Zambia between 1992 and 2018. Furthermore, the findings of this study have the potential to inform strengthening of FP strategies in Zambia and other countries in SSA.

## Theoretical framework

The theoretical framework for this study adapted the Andersen Behavioural Model of Healthcare Utilisation to understand the determinants of contraceptive use transition among women in Zambia. The Anderson model explains that an individual's utilisation of health services is a function of factors that predispose the use of health service (predisposing), factors that enable the use of health service (enabling) and factors that show the need for the use of health service (need) [35]. Here, contraceptive use among women is explained based on factors classified into individual or contextual level, enabling and need factors. The individual factors are those which are peculiar to individuals, while the contextual factors are those which are identified as present in the environment external to the individual, and thus have the potential to influence an individual's health behaviour. What constitutes the contextual characteristics are similar to individual characteristics but they are measured at the community level [35]. Contextual factors have the ability to affect individual factors, which will in turn influence health outcomes, or they can directly influence health outcomes [36]. Individual and contextual-level factors operate through intermediate factors (enabling and need) to influence contraceptive use. Thus, the Andersen's theoretical framework sheds light on how individual, contextual-level, and intermediate factors are interrelated in influencing the use of contraceptive methods among women. The framework shows how these factors play important roles in the decision to use contraceptives or not (Fig 1). Prior studies have shown how several of these factors at different levels play significant roles in determining the use of contraceptives among women of reproductive age in sub-Saharan Africa and elsewhere [34, 36–39].

## Methods and data

### Data source

This study used data from the six Zambian Demographic and Health Surveys (ZDHSs) conducted during the period 1992 and 2018. Specifically, the woman datasets (IR recode) containing data on reproductive-aged women (15–49 years) were utilised for the analysis. These ZDHSs were nationally representative surveys conducted in all the regions (provinces). The DHSs used a two-stage stratified cluster sampling design to select households by separating

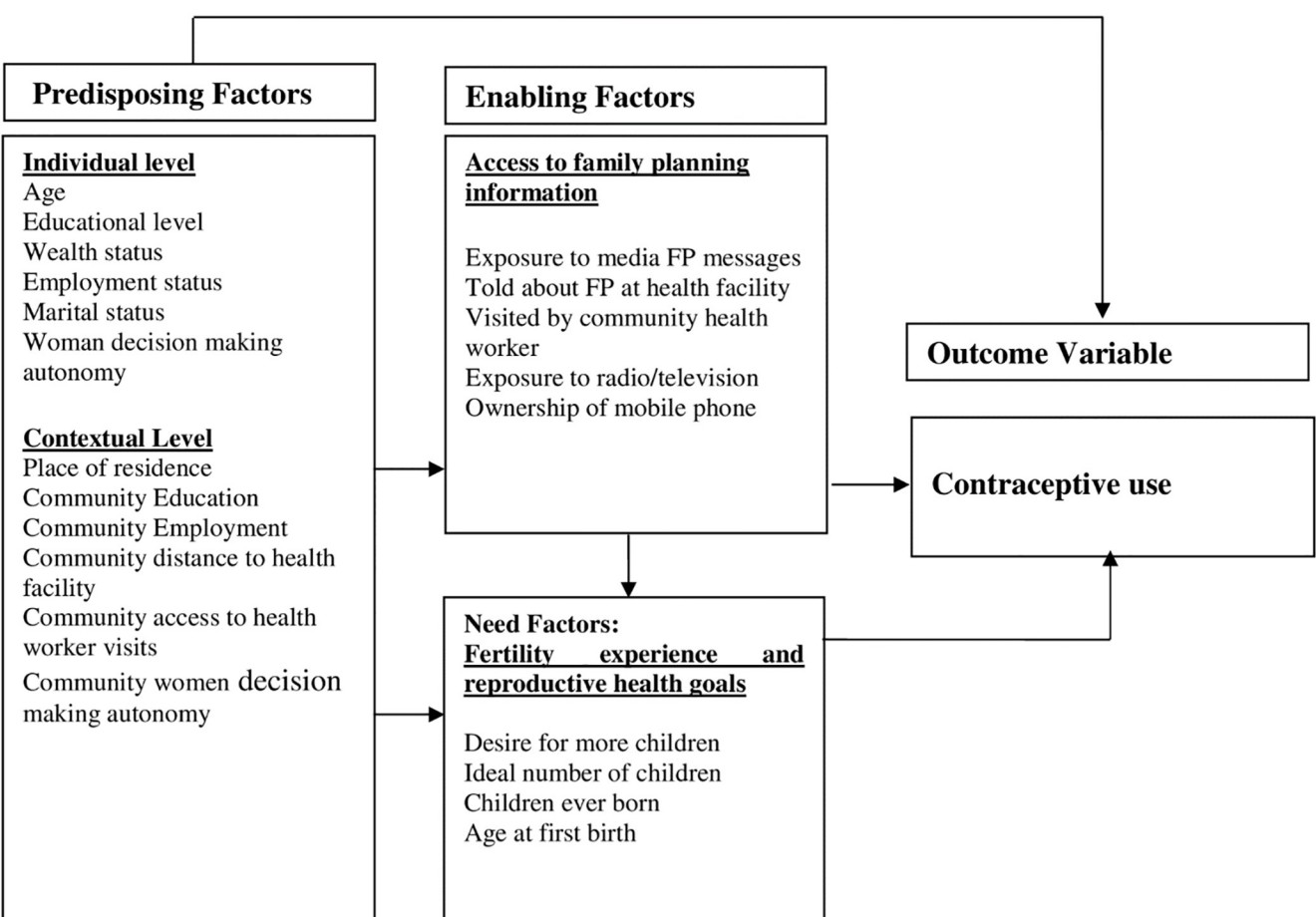

**Fig 1. Conceptual framework showing the determinants of contraceptive use at both individual and community-level adapted and modified from Andersen's behavioural model of health service utilisation.**

each region into rural and urban areas. In the first stage, enumeration areas (EAs) were selected with propositional allocation to size of the region. In the second stage, household listing operations were performed in all selected clusters. On average, 25 to 30 households per cluster [20, 40]. We extracted relevant factors for this study from the women recode (IR file) dataset. A total of samples of fecund sexually active women aged 15–49 years, 5,090 in 1992; 5,827 in 1992; 5,736 in 2001; 5,252 in 2007; 12,419 in 2013, and 10,438 in 2018 (Fig 2). Detailed sampling procedures are described in the ZDHS report [20] and elsewhere [40]. The description of the analytical sample derivation procedure is outlined in Fig 2.

### Study variables and measurement

**Dependent variable.** The outcome variable for this study is current contraceptive use. All sexually active women in the DHS were asked a question "Are you currently using any contraceptive method to prevent a pregnancy". The variable in the DHS was classified using four response categories; (i) 'no method', (ii) 'folkloric method' (iii) 'traditional method' (iv) 'modern method'. The description of the specific contraceptive methods have been described in detail elsewhere [40]. This analysis excluded women who were not sexually active or pregnant or declared infecund at the time of the survey. A binary outcome variables was then created

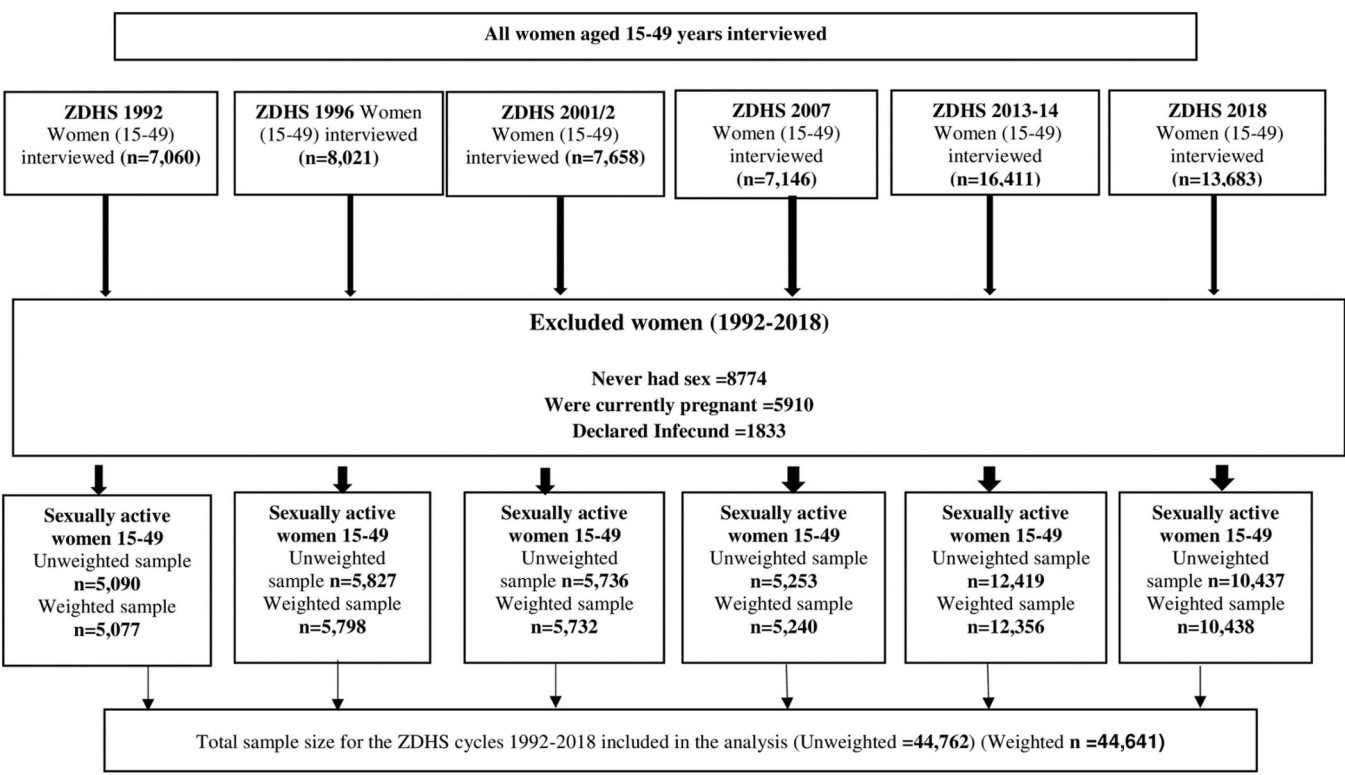

**Fig 2. Sample derivation of sexually active women from the 1992–2018 Zambia demographic and health surveys.**

from the initial variable with the classification "0" representing not using a contraceptive method and "1" representing women using any contraceptive method (folkloric/traditional or modern method).

**Independent variables.** The independent variables classified in the study were selected based on their relevance in the previous studies as predisposing or enabling factors influencing contraceptive uptake among women in SSA [32, 41–45]. In the first category, predisposing variables are socio-demographic and socio-cultural characteristics of the respondents that exist before their contraceptive behaviour. The predisposing factors are maternal age, number of living children a woman has, women's occupational status, household wealth index and a women's educational level. In the second category, enabling factors reflect the means or facilitators to access family planning services like visiting the health facility, access to FP messages, access to media, and visited by to community health workers. In the third section, needing factors are described. These are the immediate causes to use family planning services and reflect the perceived health status of the woman. Ideal family size, desire for more children and experience of child mortality are considered as needing factors.

**Community-level variables.** The community level in this analysis was defined at the EA level. This is because EA was the lowest primary sampling unit or point of data collection in the ZDHS. An EA usually contains households with similar characteristics. Thus, making analysis of community effects on contraceptive use transition appropriate. Many studies conducted in SSA have shown that region of residence, type of residence, community education, community employment, community wealth, community fertility preference, community access to media FP messages were among community level variables that have been found to be

associated with contraception use [46–49]. Apart from region and place of residence, the construction of other community-level variables from individual-level variables involved aggregating data from individual respondents within each defined DHS EA. For easier interpretation, proportions for categorization were applied on each community level variable, that is low, moderate, and high. Community-level factors considered in this study include place of residence, region, community education, community employment status, community fertility desire and community access to FP messages.

## Statistical analysis

Statistical analysis in this study included descriptive, bivariate and multivariable decomposition analysis of the transition in contraceptive use among sexually active women. The first level analysis involved conducting trend analysis on the six ZDHS datasets to measure change between the previous and the successive DHS phase. Further, the decomposition analysis of contraceptive use transition was measured between the first (1992) and the last DHS phase (2018) to examine trend change. The trend was assessed using descriptive analysis stratified by various selected predictor variables and examined separately for each phase. Two component Blinder-Oaxaca multivariable decomposition analysis was used to identify the determinants of transition in contraceptive use between 1992 and 2018. This analysis focused on how contraceptive use prevalence responds to differences in selected women's characteristics and how these variables shape the differences across the two survey points. Decomposition analysis aimed to identify the potential sources of variations in the prevalence of contraceptive use in Zambia, over the 26 years' periods. Multivariable decomposition analysis for the non-linear response model used the output from a logistic regression model since it is a dichotomous variable. The difference in the percentage of contraceptive over time is attributable to the compositional change between the two surveys and the difference in the effects of those selected independent variables. That means the change in contraceptive use is divided into the differences in characteristics (endowment component) and the effect of the selected variables (coefficient component). Besides, the Variance inflation factor (VIF) was applied to check multicollinearity among independent variables. Therefore, there was no multicollinearity between independent variables since the VIF value were less than 5.

## Ethical approval

The study did not require ethical review because the data utilised in the analysis originated from secondary sources that were already in the public domain. However, all required steps and instructions were performed in order to access the DHS program's datasets. Before the start of the data gathering operations in Zambia, the DHS standards made sure that all ethical procedures were followed. All participants who were enrolled in the DHS and older than 18 years old had to give their consent before the interview. Additionally, prior to asking the legal minors for their consent, the parents or guardians of all participants aged 15 to 17 gave their consent.

## Results

### Description of the study sample

The percent distribution of sample characteristics of all sexually active respondents captured in all the six DHSs are presented in Table 1. In the first three surveys 1992, 1996 and 2001, most of the women captured were in the age group of 15–24 years (41.6%, 41.4% and 38.7%, respectively). The 2007, 2013 and 2018 surveys had most women interviewed in the age group

**Table 1. Percent distribution of selected background characteristics of sexually active women (15–49 years) DHS 1992–2018, Zambia.**

| Background Characteristics | DHS 1992 | DHS 1996 | DHS 2001 | DHS 2007 | DHS 2013 | DHS 2018 |
|---|---|---|---|---|---|---|
| | N = 5,090 | N = 5,827 | N = 5,736 | N = 5,253 | N = 12,419 | N = 10,437 |
| **Age** | | | | | | |
| 15–24 | 2113 (41.6) | 2402 (41.4) | 2217 (38.7) | 1696 (32.3) | 3897 (31.5) | 3447 (33.0) |
| 25–34 | 1714 (33.8) | 1953 (33.7) | 1952 (34.0) | 2036 (38.9) | 4566 (37.0) | 3581 (34.3) |
| 35–49 | 1250 (24.6) | 1443 (24.9) | 1564 (27.3) | 1508 (28.8) | 3895 (32.5) | 3410 (32.7) |
| **Residence** | | | | | | |
| Urban | 2553 (51.1) | 2593 (44.0) | 2293 (40.0) | 2133 (40.7) | 5515 (44.6) | 4814 (46.1) |
| Rural | 2484 (48.9) | 3245 (56.0) | 3440 (60.0) | 3107 (59.3) | 6841 (55.4) | 5625 (53.9) |
| **Education level** | | | | | | |
| None | 872 (17.2) | 792 (13.7) | 746 (13.0) | 605 (11.6) | 1126 (9.1) | 855 (8.2) |
| Primary | 3034 (59.8) | 3427 (59.1) | 3356 (58.5) | 2930 (56.0) | 6072 (49.2) | 4719 (45.2) |
| Secondary | 1075 (21.2) | 1402 (24.2) | 1449 (25.3) | 1414 (26.0) | 4498 (36.4) | 4248 (40.7) |
| Higher | 94 (1.9) | 176 (3.0) | 181 (3.2) | 290 (5.4) | 650 (5.3) | 617 (5.9) |
| **Number of living children** | | | | | | |
| 0–1 | 2020 (39.3) | 2252 (38.8) | 2104 (36.7) | 1664 (31.8) | 3761 (30.4) | 3505 (33.6) |
| 2–3 | 1288 (25.4) | 1637 (28.2) | 1709 (29.8) | 1658 (31.7) | 3777 (30.6) | 3192 (30.6) |
| 4–5 | 842 (17.0) | 1023 (17.7) | 1003 (17.5) | 1087 (20.8) | 2715 (22.0) | 2155 (20.6) |
| 6+ | 927 (18.3) | 886 (15.3) | 917 (16.0) | 830 (15.7) | 2103 (17.0) | 1587 (15.2) |
| **Wealth status** | | | | | | |
| Poor | _ | 2242 (38.7) | 2120 (37.0) | 1920 (36.6) | 4509 (36.5) | 3753 (36.0) |
| moderate | _ | 1028 (17.7) | 1151 (20.0) | 956 (18.2) | 2421 (19.3) | 1922 (18.4) |
| Rich | _ | 2527 (43.6) | 2462 (43.0) | 2365 (45.2) | 5426 (44.2) | 4764 (45.6) |
| **Employment status** | | | | | | |
| Unemployed | 2398 (47.2) | 2917 (50.3) | 2417 (42.2) | 2464 (47.1) | 5550 (45.1) | 5146 (49.3) |
| Employed | 2678 (52.8) | 2877 (49.7) | 3312 (57.8) | 2768 (52.9) | 6754 (54.9) | 52 93 (50.7) |
| **Experienced child mortality** | | | | | | |
| No | 4096 (80.7) | 4701 (81.1) | 4791 (83.6) | 4516 (86.2) | 1 (87.5) | 9036 (86.6) |
| Yes | 980 (19.3) | 1097 (18.9) | 941 (16.4) | 724 (13.8) | 1545 (12.5) | 1402 (13.4) |
| **Fertility intention** | | | | | | |
| Want another | 2453 (71.5) | 3976 (68.6) | 3616 (63.3) | 3013 (57.6) | 7514 (61.0) | 6192 (59.3) |
| Want no more | 829 (24.2) | 1622 (28.0) | 1960 (34.3) | 1841 (35.2) | 4173 (33.9) | 3655 (35.0) |
| Undecided | 148 (4.3) | 197 (3.4) | 137 (2.4) | 377 (7.2) | 628 (5.1) | 592 (5.7) |
| **Ideal number of children** | | | | | | |
| 0–3 | 507 (10.0) | 819 (14.1) | 1294 (22.6) | 1154 (22.2) | 2601 (21.5) | 2364 (22.6) |
| 4–5 | 1814 (35.7) | 2334 (40.3) | 2409 (42.0) | 2247 (42.9) | 5556 (45.0) | 4862 (46.6) |
| 6+ | 2757 (54.3) | 2645 (45.6) | 2029 (35.4) | 1838 (35.1) | 4199 (34.0) | 3213 (30.8) |
| **Exposure to media FP messages** | | | | | | |
| No | 4734 (93.2) | 4289 (74.0) | 4175 (72.8) | 4002 (76.0) | 9589 (77.6) | 8882 (85.1) |
| Yes | 343 (6.8) | 1509 (26.0) | 1557 (27.2) | 1238 (24.0) | 2767 (22.4) | 1557 (14.9) |

Note: (-) means data for the variable was not collected in the respective DHS year

25–34 years (38.9%, 37.0% and 34.3% respectively. In terms of education level, most of the sampled women had primary level of education, ranging from 59.8% in 1992 to 45.2% in 2018. Those with tertiary education increased from 1.9% in 1992 to 5.9% in 2018. Over one third of the respondents in all the survey years were from poor households, while those from rich households increased from 43.6% in 1996 to 45.6% in 2018.

In all the surveys except for 1996, women who were in employment were in majority compared to those not in employment. The proportion of those employed has been fluctuating with the highest being 2001, 57.8% and lowest was 49.7% in 1996. Most of the respondents desired 6 or more children in the first two survey years (1992 and 1996). However, the percentage of women desiring 6 or children reduced during the period of analysis from 54.3% in 1992 to 30.8% in 2018. Further, exposure to media FP messages increased from 6.8% in 1992 to 14.9% in 2018 (Table 1).

## Contraceptive use transition in Zambia

Overall, the findings show that during the period 1992 to 2018, contraceptive use in Zambia increased from 14.2% to 45.0%. In urban areas, the prevalence increased from 17.5% in 1992 to 45.6% in 2018. On the other hand, in rural areas, the increase was from 10.8% in 1992 to 44.5% 2018 (Fig 3). The biggest contribution to change in contraceptive use between 1992 and 2018 in Zambia occurred in rural areas (33.7%). Contraceptive use change associated with urban areas during the same period was 28.1%. At national level, the biggest amount of change happened between 1992 and 1996 (10.3%). The greatest amount of change in rural areas happened during the period 2001 to 2007 (10.8%). On the other hand, the largest change in urban areas took place during the period 1992 to 1996 (12.9%) (Fig 4).

Trend in contraceptive use by method type showed that the most increase was recorded for injections which increased from 1.1% in 1992 to 52.5% in 2018. Similarly, Utilisation of implants increased during the period 2007 to 2018 (from 1.1% to 17.7%). On the other hand, the usage of the Pill declined from 43.1% in 1992 to 14.3% in 2018. Traditional methods also recorded reduction from 6.4% to 1.7% during the analysis period 1992 to 2018 (Fig 5).

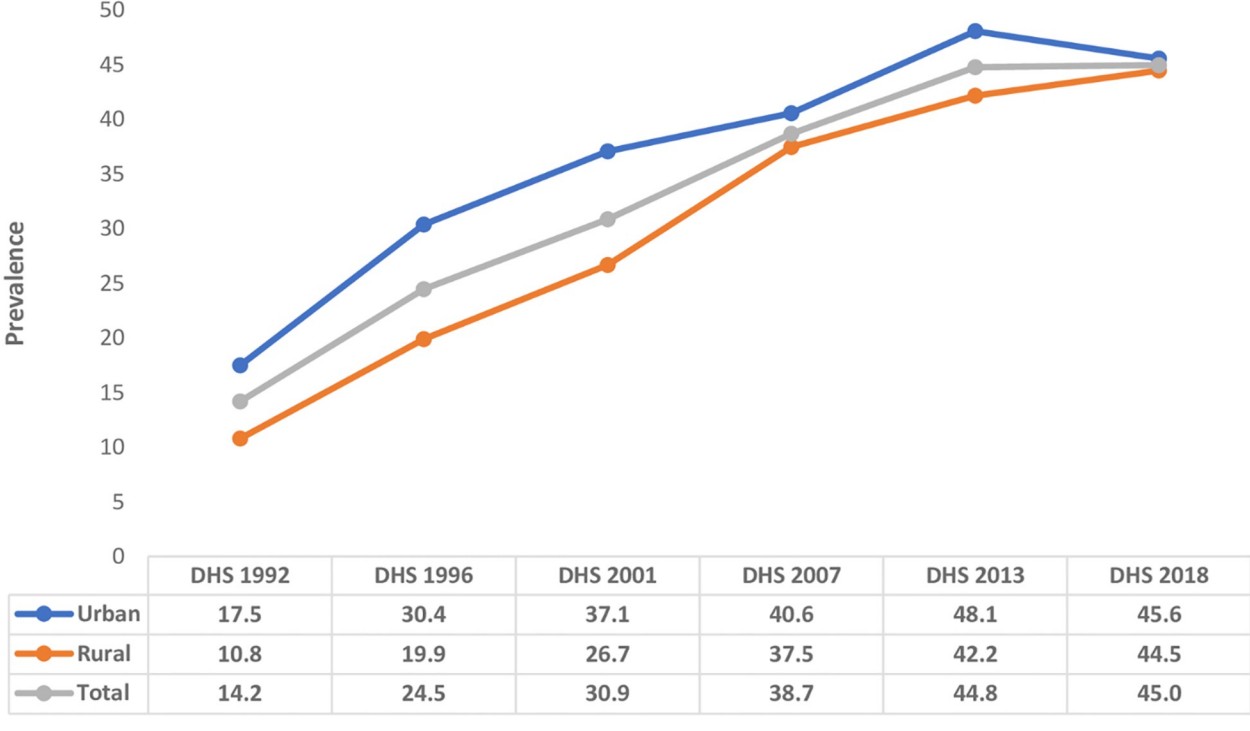

| | DHS 1992 | DHS 1996 | DHS 2001 | DHS 2007 | DHS 2013 | DHS 2018 |
|---|---|---|---|---|---|---|
| Urban | 17.5 | 30.4 | 37.1 | 40.6 | 48.1 | 45.6 |
| Rural | 10.8 | 19.9 | 26.7 | 37.5 | 42.2 | 44.5 |
| Total | 14.2 | 24.5 | 30.9 | 38.7 | 44.8 | 45.0 |

**Fig 3. Trend in contraceptive use prevalence among sexually active women 15–49 years by residence, Zambia, DHS 1992–2018.**

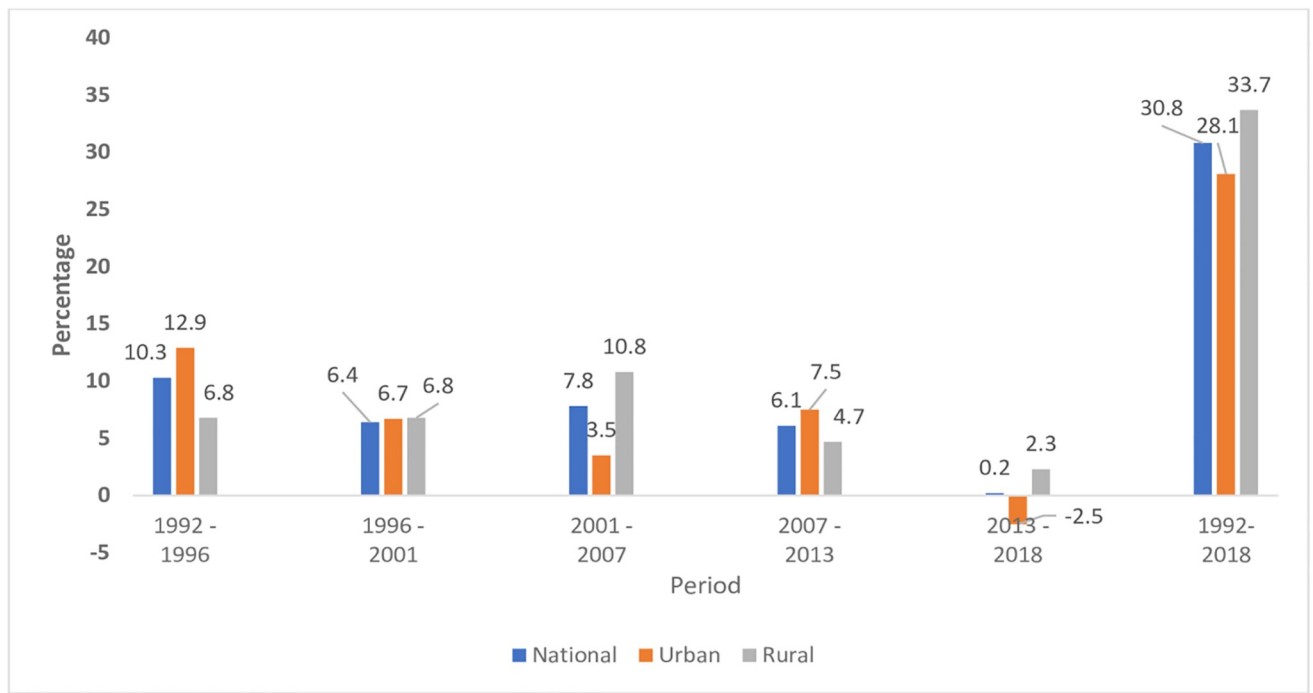

**Fig 4. Trend in rate of change in contraceptive use among sexually active women, Zambia, DHS 1992–2018.**

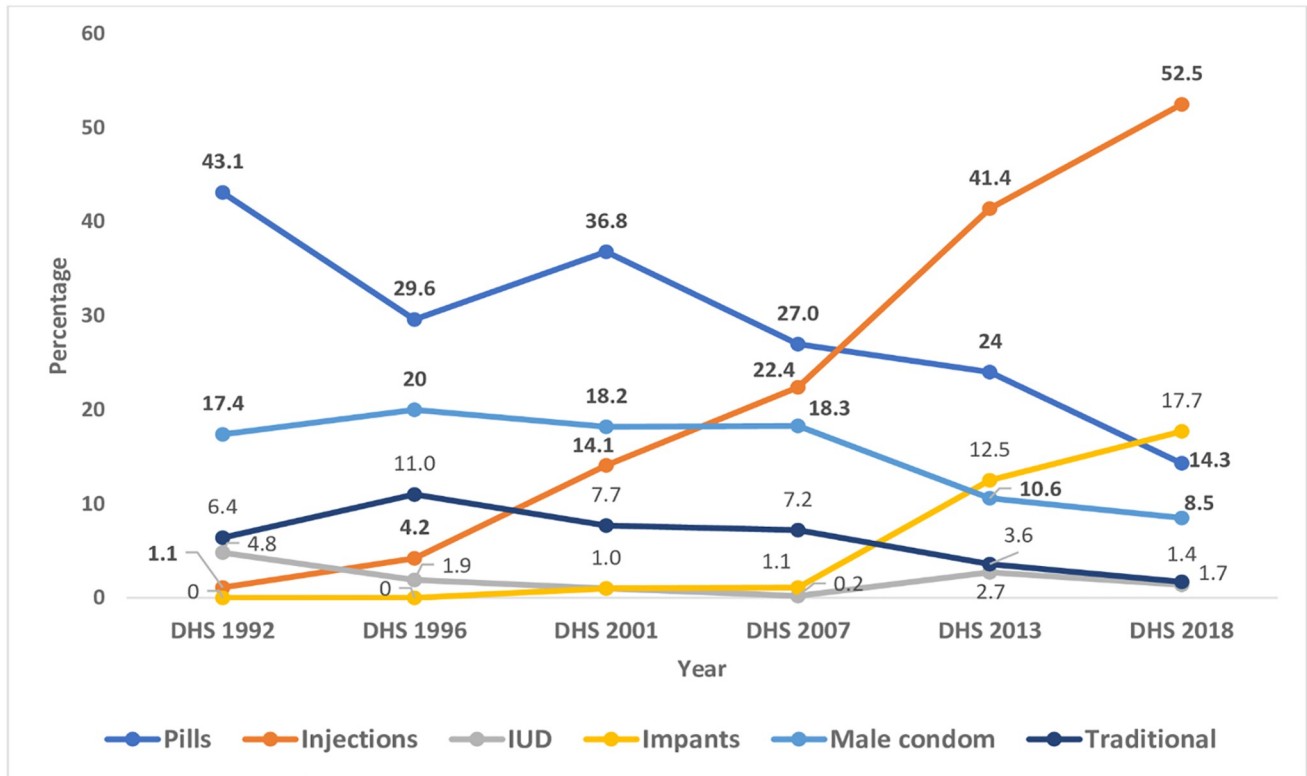

**Fig 5. Trends in type of contraceptive method use among sexually active women in Zambia DHS 1992–2018.**

## Description of contraceptive use transition by socio-demographic characteristics of sexually active women

Trend change in contraceptive use among sexually active reproductive aged women in Zambia varied with background during the period 1992 to 2018. The prevalence of contraceptive use increased in most of the categories that were analysed in this study (Table 2). Overall, the prevalence of contraceptive use in Zambia increased by 30.8% among sexually active women during the period 1992 to 2018. Based on age, the most increase in contraceptive use was observed among women aged 25–34 years, 35.2 percentage points in the same period 1992 to 2018. In terms of residence, women living in rural areas showed the most improvement in percentage change of contraceptive use (33.7% compared to 28.1% in urban areas). Regional variations were observed in terms of contraceptive use change over time in Zambia. The most increase in contraceptive by region during the period 1992 to 2018 was observed in Eastern province (42.5%) and the lowest increase was observed among women in Western province (18.3%).

In terms of education, those with a primary level of education recoded the most increase in utilisation of contraceptive methods (34.8.0%) while those with tertiary education recoded a minor drop in use of contraceptive (-6.3%). Women who were in employment recorded an increase of 28.4% in prevalence of contraceptive use. According to the number of living children, women who had 2–3 children recorded the highest increase in percentage of contraceptive use (36.2%) and the lowest increase was observed among those with 0–1 child (22.0%). Women who had 4–5 children ever born showed the highest increase in utilisation of contraceptive methods among women (25.1%). On the other hand, those with 0–1 child showed the least increase (14.4%).

In terms of exposure to radio and media FP messages, women who had expose to radio recorded an increase of 19.5% in contraceptive use while those with exposure to media FP messages recorded an increase of 18.3% between 1992 and 2018. On the other hand, women who did not experience child mortality saw an improvement of 33.3% in the utilisation of contraceptive use during the analysis period. Similarly, the women from communities with high education recorded an increase of 30.3% in the uptake of contraception (Table 2).

## Determinants of contraceptive use transition in Zambia, 1992–2018

Table 3 presents decomposition analysis results of contraceptive use transition in Zambia during the period 1992–2018. For endowments or compositional effect, a positive coefficient shows that the increase in the proportion of women possessing that characteristic between 1992 and 2018 was associated with an increase in the utilisation of contraceptive methods. A negative coefficient indicates that if the percentage of women with that attribute had remained constant between 1992 and 2018, the prevalence of contraception use in 2018 would have been higher.

Results show that overall, both changes in compositional structure of women and changes in women's contraceptive behaviour significantly contributed to the increase in the prevalence of contraceptive use in Zambia. The decomposition results showed that the increase in contraceptive use over time was explained by the difference in the selected women's characteristics and contraceptive behavioural changes between the two survey points. About 15% of the increase in the prevalence of contraceptive use was due to the differences in the composition of women's characteristics. On the other hand, the change due to the differences in the effect of the selected variables was 85% (Table 3).

Multivariable decomposition analysis of the determinants of transition in contraceptive use has revealed that between 1992 and 2018: maternal education, region, number of living children, fertility preference, exposure to watching television, experience of child mortality,

**Table 2. Percent distribution of contraceptive use among sexually active women (15–49 years) by background characteristics, DHS 1992–2018, Zambia.**

| Background Characteristics | DHS 1992 (N = 5,090) | DHS 1996 (N = 5,827) | DHS 2001 (N = 5,736) | DHS 2007 (N = 5,253) | DHS 2013 (N = 12,419) | DHS 2018 (N = 10,437) | Percentage point change in use of contraceptives 1992 to 2018 |
|---|---|---|---|---|---|---|---|
| | N (%) | N (%) | N (%) | N (%) | N (%) | N (%) | |
| **Age** | *** | *** | *** | *** | *** | *** | |
| 15–24 | 230 (10.9) | 518 (21.6) | 598 (27.0) | 607 (35.8) | 1422 (36.5) | 1337 (38.8) | 27.9 |
| 25–34 | 321 (18.8) | 580 (29.7) | 793 (40.7) | 934 (45.9) | 2478 (54.3) | 1934 (54.0) | 35.2 |
| 35–49 | 169 (13.5) | 321 (22.3) | 378 (24.2) | 488 (32.4) | 1638 (42.1) | 1426 (41.8) | 27.5 |
| **Residence** | *** | *** | *** | | *** | | |
| Urban | 453 (17.5) | 775 (30.4) | 850 (37.1) | 865 (40.6) | 2654 (48.1) | 2196 (45.6) | 28.1 |
| Rural | 268 (10.8) | 644 (19.9) | 919 (26.7) | 1164 (37.5) | 2884 (42.2) | 2501 (44.5) | 33.7 |
| **Province** | *** | *** | *** | *** | *** | *** | |
| Central | 43 (9.4) | 85 (17.9) | 95 (23.4) | 156 (31.7) | 448 (39.9) | 409 (46.6 | 37.2 |
| Copperbelt | 196 (16.2) | 272 (25.0) | 400 (36.4) | 360 (40.9) | 910 (46.3) | 728 (44.8) | 28.6 |
| Eastern | 56 (10.1) | 173 (21.3) | 191 (27.0) | 395 (53.4) | 746 (50.0) | 657 (52.6) | 42.5 |
| Luapula | 27 (9.3) | 45 (8.9) | 134 (30.8) | 60 (16.1) | 290 (33.4) | 289 (36.2) | 26.9 |
| Lusaka | 183 (20) | 350 (34.4) | 368 (41.6) | 334 (39.4) | 1202 (50.2) | 1006 (47.6) | 27.6 |
| Muchinga | - | - | - | - | 271 (43.1) | 297 (52.8) | 52.8 |
| Northern | 74 (17.5) | 183 (30.1) | 212 (28.8) | 252 (36.0) | 411 (45.2) | 322 (40.9) | 23.4 |
| North-western | 12 (8.9) | 85 (40.6) | 70 (25.9) | 90 (33.8) | 192 (34.3) | 234 (42.9) | 34.0 |
| Southern | 74 (9.6) | 141 (23.2) | 189 (29.5) | 248 (45.1) | 797 (50.6) | 526 (42.9) | 33.3 |
| Western | 55 (16.6) | 85 (17.8) | 111 (20.1) | 134 (34.4) | 274 (32.4) | 231 (34.9) | 18.3 |
| **Education level** | *** | *** | | * | *** | *** | |
| None | 68 (7.8) | 131 (16.6) | 160 (21.4) | 204 (33.7) | 412 (36.6) | 305 (35.7) | 27.9 |
| Primary | 367 (12.1) | 769 (22.4) | 960 (28.6) | 1136 (38.8) | 2734 (45.0) | 2215 (46.9) | 34.8 |
| Secondary | 239 (22.2) | 422 (30.1) | 558 (38.5) | 564 (39.9) | 2058 (45.8) | 1916 (45.1) | 22.9 |
| Higher | 46 (48.6) | 97 (55.0) | 91 (50.4) | 126 (43.3) | 334 (51.4) | 261 (42.3) | -6.3 |
| **Wealth status** | | *** | *** | *** | *** | ** | |
| Poor | - | 445 (19.9) | 476 (22.4) | 735 (38.3) | 1781 (39.6) | 1595 (42.5) | 22.3 |
| Moderate | - | 198 (19.3) | 339 (29.4) | 314 (32.8) | 1140 (47.1) | 926 (48.2) | 28.9 |
| Rich | - | 776 (30.7) | 955 (38.8) | 981 (41.5) | 2618 (48.3) | 2177 (45.7) | 15.0 |
| **Employment status** | | | | | | | |
| Unemployed | 227 (9.5) | 641 (22.0) | 710 (29.4) | 942 (38.2) | 2448 (44.10) | 2219 (43.1) | 33.6 |
| Employed | 493 (18.4) | 777 (27.0) | 1058 (31.9) | 1084 (39.2) | 3066 (45.4) | 2478 (46.8) | 28.4 |
| **Living children** | *** | *** | *** | *** | *** | *** | |
| 0–1 | 163 (8.1) | 357 (15.9) | 439 (20.9) | 438 (26.3) | 1050 (29.9) | 1052 (30.0) | 22.0 |
| 2–3 | 228 (17.7) | 505 (30.9) | 616 (36.1) | 766 (46.2) | 2039 (54.0) | 1720 (53.9) | 36.2 |
| 4–5 | 166 (19.7) | 281 (27.5) | 411 (41.0) | 479 (44.1) | 1427 (52.6) | 1160 (53.8) | 34.1 |
| 6+ | 164 (17.7) | 276 (31.1) | 303 (33.1) | 346 (41.7) | 1022 (48.6) | 765 (48.2) | 30.5 |
| **Children ever born** | *** | *** | *** | *** | *** | *** | |
| 0–1 | 138 (7.9) | 305 (15.6) | 381 (20.9) | 381 (25.8) | 960 (27.7) | 987 (30.0) | 14.4 |
| 2–3 | 214 (17.4) | 447 (30.0) | 550 (35.3) | 676 (46.3) | 1920 (54.1) | 1643 (53.7) | 23.7 |
| 4–5 | 145 (17.6) | 294 (29.1) | 412 (39.8) | 512 (45.4) | 1345 (52.4) | 1131 (54.2) | 25.1 |
| 6+ | 223 (17.4) | 373 (27.6) | 427 (32.4) | 460 (39.2) | 1314 (47.4) | 936 (46.9) | 19.3 |
| **Ideal Number of children** | *** | *** | *** | | ** | *** | |
| 0–3 | 104 (20.6) | 224 (27.3) | 415 (32.1) | 444 (38.5) | 1092 (42.0) | 940 (39.8) | 19.2 |
| 4–5 | 293 (16.1) | 631 (27.0) | 820 (34.1) | 868 (38.6) | 2609 (47.0) | 2330 (47.9) | 31.8 |
| 6+ | 323 (11.7) | 565 (21.3) | 534 (26.3) | 718 (39.1) | 1837 (43.7) | 1427 (44.4) | 32.7 |
| **Exposure to radio** | *** | *** | *** | *** | *** | *** | |

*(Continued)*

**Table 2.** (Continued)

| Background Characteristics | DHS 1992 (N = 5,090) | DHS 1996 (N = 5,827) | DHS 2001 (N = 5,736) | DHS 2007 (N = 5,253) | DHS 2013 (N = 12,419) | DHS 2018 (N = 10,437) | Percentage point change in use of contraceptives 1992 to 2018 |
|---|---|---|---|---|---|---|---|
| | N (%) | N (%) | N (%) | N (%) | N (%) | N (%) | |
| No | 209 (9.5) | 496 (19.4) | 616 (25.2) | 547 (34.5) | 1954 (41.2) | 2273 (42.2) | 22.8 |
| Yes | 509 (17.7) | 920 (28.5) | 1153 (35.1) | 1480 (40.5) | 3582 (47.1) | 2424 (48.0) | 19.5 |
| **Exposure to television** | *** | *** | *** | * | *** | | |
| No | 486 (12.0) | 879 (20.9) | 1079 (26.7) | 1277 (37.5) | 2957 (42.3) | 2675 (44.5) | 23.6 |
| Yes | 234 (23.0) | 540 (34.1) | 688 (40.2) | 751 (41.0) | 2580 (48.2) | 2022 (45.7) | 22.7 |
| **Exposure to newspaper** | *** | *** | *** | *** | | | |
| No | 320 (10.7) | 937 (21.3) | 1265 (28.5) | 1294 (36.8) | 3642 (44.6) | 3698 (44.7) | 23.4 |
| Yes | 397 (19.2) | 480 (34.6) | 502 (39.3) | 734 (42.8) | 1883 (45.4) | 999 (46.2) | 11.6 |
| **Exposure to media FP messages** | *** | *** | *** | ** | *** | | |
| No | 630 (13.3) | 894 (20.9) | 1142 (27.4) | 1504 (37.6) | 4146 (43.2) | 4002 (45.1) | 31.8 |
| Yes | 90 (26.3) | 525 (34.8) | 627 (40.3) | 526 (42.4) | 1393 (50.3) | 695 (44.6) | 18.3 |
| **Fertility preference** | ** | | | | *** | ** | |
| Want another child | 372 (15.1) | 931 (23.4) | 1067 (29.5) | 1176 (39.0) | 3230 (43.0) | 2719 (43.9) | 28.8 |
| Want no more children | 181 (21.8) | 453 (27.9) | 669 (34.2) | 708 (38.5) | 2025 (48.5) | 1733 (47.4) | 25.6 |
| Undecided | 26 (17.7) | 35 (17.8) | 29 (21.2) | 141 (37.3) | 268 (42.6) | 245 (41.5) | 23.8 |
| **Experienced child mortality** | *** | *** | *** | *** | *** | *** | |
| No | 681 (16.6) | 1299 (27.6) | 1652 (34.5) | 1906 (42.2) | 5372 (49.7) | 4512 (49.9) | 33.3 |
| Yes | 39 (4.0) | 120 (10.9) | 117 (12.4) | 123 (17.0) | 167 (10.8) | 184 (13.2) | 9.2 |
| **Community education** | | * | | | * | | |
| Low | 357 (12.8) | 769 (22.5) | 846 (28.5) | 1031 (38.2) | 2471 (43.3) | 1976 (44.0) | 31.2 |
| Medium | 257 (15.6) | 395 (27.8) | 542 (31.5) | 586 (40.3) | 1190 (47.7) | 1068 (43.7) | 28.1 |
| High | 106 (16.9) | 256 (26.8) | 381 (36.6) | 412 (38.1) | 1878 (45.2) | 1653 (47.2 | 30.3 |
| **Community wealth** | | | | | | | |
| Low | _ | 712 (22.7) | 741 (26.8) | 1010 (39.3) | 2539 (43.1) | 1982 (44.2) | 21.4 |
| Medium | _ | 362 (26.0) | 564 (34.3) | 525 (38.1) | 1210 (46.4) | 1187 (43.5) | 17.5 |
| High | _ | 346 (27.2) | 465 (35.0) | 495 (38.4) | 1790 (46.3) | 1527 (47.4) | 20.2 |
| **Community employment** | *** | | | | *** | | |
| Low | 301 (11.5) | 804 (24.7) | 977 (33.22) | 1156 (40.7) | 2942 (47.4) | 2359 (45.8) | 34.3 |
| Medium | 262 (17.8) | 347 (23.6) | 469 (29.3) | 495 (36.4) | 1142 (42.4) | 1159 (44.8) | 27.0 |
| High | 157 (15.9) | 268 (24.9) | 324 (27.1) | 379 (36.3) | 1454 (42.1) | 1179 (43.6) | 27.7 |
| **Community family size desire** | * | ** | * | | ** | | |
| Low | 351 (16.6) | 714 (27.5) | 974 (33.1) | 1067 (38.7) | 3178 (46.11) | 2711 (46.1) | 29.5 |
| Medium | 204 (12.9) | 348 (23.2) | 446 (29.9) | 528 (41.5) | 1251 (46.0) | 1088 (44.4) | 31.5 |
| High | 165 (12.0) | 358 (21.0) | 350 (27.0) | 434 (35.9) | 1110 (40.5) | 898 (42.6) | 30.6 |
| **Community early childbearing** | | | | * | | | |
| Low | 340 (15.9) | 648 (25.3) | 840 (30.4) | 1015 (39.3) | 3085 (45.8) | 2607 (45.9) | 30.0 |
| Medium | 201 (12.2) | 391 (23.6) | 482 (30.6) | 633 (41.6) | 1067 (42.9) | 1008 (45.7) | 33.5 |
| High | 179 (13.9) | 380 (24.1) | 448 (32.1) | 382 (33.6) | 1387 (44.3) | 1082 (42.4) | 28.5 |
| **Community exposure to FP messages** | * | * | | | | | |
| Low | 333 (12.8) | 605 (22.0) | 661 (25.9) | 910 (38.8) | 2811 (44.0) | 2238 (44.9) | 32.1 |

*(Continued)*

**Table 2.** (Continued)

| Background Characteristics | DHS 1992 (N = 5,090) | DHS 1996 (N = 5,827) | DHS 2001 (N = 5,736) | DHS 2007 (N = 5,253) | DHS 2013 (N = 12,419) | DHS 2018 (N = 10,437) | Percentage point change in use of contraceptives 1992 to 2018 |
|---|---|---|---|---|---|---|---|
| | N (%) | N (%) | N (%) | N (%) | N (%) | N (%) | |
| Medium | 218 (14.0) | 437 (26.3) | 467 (31.0) | 498 (38.9) | 1115 (48.1) | 1080 (44.1) | 30.1 |
| High | 169 (18.4) | 377 (27.4) | 642 (38.3) | 622 (38.5) | 1612 (44.3) | 1379 (46.0) | 27.6 |
| **Total** | **14.2** | **24.5** | **30.9** | **38.7** | **44.8** | **45.0** | **30.8** |

Note: (-) means data for the variable was not collected in the respective DHS year;

* = p<0.05;

** = p<0.01;

***p<0.001

community education were the major contributors to trend increase in contraceptive use among sexually active women.

The increment in the proportion of women with secondary level education showed a significant positive contribution of 5.87% to the increase in the contraceptive prevalence rate. Furthermore, an increase in the composition of women who had higher level education and those from communities with higher level education contributed about 1% and 2% to the increase in contraceptive use, respectively. A reduction in the proportion of women whose ideal number of children was 6 or more made a significant positive contribution of 5.63% to contraceptive use increase. Additionally, a reduction in the proportion of women who experienced child mortality contributed to a significant increase in the prevalence of contraceptive use by 7.71%.

## Discussion

The purpose of this study was to examine the drivers of contraceptive use transition in Zambia during the period 1992 to 2018. Increasing utilisation of contraceptive methods is critical for improving maternal health in sub-Saharan Africa [6, 50, 51]. Results from this study show that contraceptive use increased significantly in Zambia during the analysis period. The increase was mainly as a result of changes in women's contraceptive behaviour than the change in their compositional factors. The decomposition analysis show that about 15% of the overall increase in contraceptive use in Zambia was as a result of changes in women's compositional factors. On the other hand, 85% of the improvement in contraceptive use was explained by changes in women's contraceptive behaviour.

Modern contraceptive use increase in Zambia was mainly as a result of improvement in the utilisation of injections, implants and intrauterine device and women of reproductive age. Modern contraceptive methods are effective in promoting reproductive health, gender equality, and socio-economic development [52–54]. Modern contraceptives not only contribute to maternal health and reduces the risk of unintended pregnancies but also allows women to pursue education, careers, and personal goals. Additionally, modern contraceptives offer a range of options, providing women or couples with choices that align with their preferences and health considerations. Promoting the use of modern contraceptive methods is a fundamental aspect of family planning programmes and global health initiatives, contributing to the overall well-being of women and children [6, 55].

The change in women's contraceptive behaviour in Zambia may be attributed to government position to reposition family planning programming through implementation of favourable public sexual reproductive health policies and programmes. These included the

Table 3. Contribution of explanatory variables to the difference in contraceptive use among sexually active women between 1992–2018, ZDHS.

| Background Characteristics | Due to differences in characteristics (E) | | Due to differences in coefficients (C) | |
|---|---|---|---|---|
| | Coefficients | Percent | Coefficients | Percent |
| **Age** | | | | |
| 15–24 | **Ref** | | **Ref** | |
| 25–34 | -0.00037*** | -0.09 | 0.01242 | 3.12 |
| 35–49 | -0.02375*** | -5.96 | 0.01507 | 3.78 |
| **Residence** | | | | |
| Urban | -0.00183* | -0.46 | 0.00950 | 2.38 |
| Rural | **Ref** | | **Ref** | |
| **Region** | | | | |
| Central | 0.00003 | 0.01 | 0.01306** | 3.28 |
| Copperbelt | 0.00166 | 0.42 | 0.00823 | 2.06 |
| Eastern | 0.00032 | 0.08 | 0.00507 | 1.27 |
| Luapula | -0.00213*** | -0.54 | -0.00074 | -0.19 |
| Lusaka | **Ref** | | **Ref** | |
| Northern | -0.00049 | -0.12 | -0.00667 | -1.67 |
| North-western | -0.00120* | -0.30 | 0.00179 | 0.45 |
| Southern | 0.00163* | 0.41 | -0.00120 | -0.30 |
| Western | 0.00065*** | 0.16 | -0.01332*** | -3.34 |
| **Woman's education level** | | | | |
| None | **Ref** | | **Ref** | |
| Primary | -0.01104*** | -2.77 | 0.01426 | 3.58 |
| Secondary | 0.02341*** | 5.87 | -0.01127 | -2.83 |
| Higher | 0.00286* | 0.72 | -0.00563** | -1.41 |
| **Employment status** | | | | |
| No working | **Ref** | | **Ref** | |
| Working | -0.00022 | -0.05 | -0.02430* | -6.10 |
| **Number of living children** | | | | |
| 0–1 | **Ref** | | **Ref** | |
| 2–3 | 0.00683*** | 1.71 | -0.00180 | -0.45 |
| 4–5 | 0.01351*** | 3.39 | -0.00550 | -1.38 |
| 6+ | -0.00589*** | -1.48 | -0.01556 | -3.90 |
| **Ideal number of children** | | | | |
| 0–3 | **Ref** | | **Ref** | |
| 4–5 | -0.00295 | 0.07 | 0.00227 | 0.57 |
| 6+ | 0.01629** | 5.63 | 0.02040 | 5.12 |
| **Desire for more children** | | | | |
| Want another | **Ref** | | **Ref** | |
| No more | 0.00142 | 0.36 | -0.00827 | -2.08 |
| Undecided | -0.00008 | -0.02 | 0.00002* | 0.01 |
| **Exposure to listening radio** | | | | |
| No | **Ref** | | **Ref** | |
| Yes | -0.00177* | -0.45 | -0.01422 | -3.57 |
| **Exposure of watching television** | | | | |
| No | **Ref** | | **Ref** | |
| Yes | 0.00401 | 1.01 | -0.00839* | -2.10 |
| **Exposure of reading newspaper** | | | | |
| No | **Ref** | | **Ref** | |

*(Continued)*

**Table 3.** (Continued)

| Background Characteristics | Due to differences in characteristics (E) | | Due to differences in coefficients (C) | |
|---|---|---|---|---|
| | Coefficients | Percent | Coefficients | Percent |
| Yes | -0.00138 | -0.35 | -0.00844 | -2.12 |
| **Exposure to media FP messages** | | | | |
| No | Ref | | Ref | |
| Yes | 0.00053 | 0.13 | 0.00136 | -0.34 |
| **Experienced child death** | | | | |
| No | Ref | | Ref | |
| Yes | -0.00479*** | 7.71 | -0.00479 | -1.20 |
| **Community education** | | | | |
| Low | Ref | | Ref | |
| Medium | 0.00125 | 0.31 | 0.00553 | 1.39 |
| High | 0.00696 | 1.75 | 0.01097* | 2.75 |
| **Community employment status** | | | | |
| Low | Ref | | Ref | |
| Medium | -0.00012 | -0.03 | -0.01882* | -4.72 |
| High | -0.00053* | -0.13 | -0.01646** | -4.13 |
| **Community desired family size** | | | | |
| Low | Ref | | Ref | |
| Medium | -0.00012 | -0.03 | 0.00672 | 1.68 |
| High | -0.00053 | -0.13 | 0.00382 | 0.96 |
| **Community early childbearing** | | | | |
| Low | Ref | | Ref | |
| Medium | -0.00188 | -0.47 | 0.01262 | 3.17 |
| High | 0.00027* | 0.07 | -0.00752 | -1.89 |
| **Community access to media FP messages** | | | | |
| Low | Ref | | Ref | |
| Medium | 0.00070 | 0.17 | -0.01040 | -2.61 |
| High | -0.00171 | 0.43 | -0.01426* | -3.58 |
| Constant | | | 0.35078** | 88.01 |
| Total | 0.05921*** | 14.85 | 0.33936*** | 85.15 |

* = p<0.05;

** = p<0.01;

***p<0.001; Ref = Reference Category

Reproductive Health Policy launched in the year 2000, Family Planning Guidelines and Protocols developed in 2006; the Integrated Family Planning Scale-Up Plan 2013–2020 and the Adolescent Health Strategy 2017–2021 [23, 26, 56, 57].

These strategies could have collectively strengthened family planning programmes through improved financial commitment to FP programming, which improved procurement of contraceptive commodities and addressed bottlenecks in the supply chain. This led to increased access to contraceptives among women in Zambia [23]. Furthermore, family planning programmes increased community awareness on the benefits of modern contraceptives use and increased mass-media dissemination of information regarding sources of family planning services [27, 29, 31, 58, 59]. The study finding has significant policy implication for strengthening FP policy and programmes in order to achieve the FP2030 targets following failing to achieve the FP2020 target of modern contraceptive prevalence rate of 58%.

Consistent with prior studies conducted in Rwanda, Tanzania, Ethiopia, and Kenya [12, 60–62]. This study showed that change in contraceptive use was mainly as a result of changes in women's contraceptive behaviour. However, the finding that 15% of the increase in contraceptive use in Zambia was explained by differences in women's characteristics is lower compared to what was found in Ethiopia (34%), Cameroon (69%) and Rwanda (23%) [12, 17, 61]. The variations in population structure and socio-economic characteristics of women in the study countries may account for the disparity in the observed changes in CPR across different counties. Despite the country differences, the finding of this study imply that when the population structure of women changes in accordance with key determinants, a considerable change in CPR occurs.

Among the key factors influencing the rise in using of contraceptive methods in Zambia were compositional changes relating women's education (secondary or higher). Because of the apparent influence of women's education on contraception behaviour and fertility, the proportion of women with secondary education increased in the country during the analysis period and was accompanied by an increase in the usage of contraceptives over time. The improvement in the proportion of women attaining secondary education in Zambia could be attributed to the country's implementation of basic education policy, which enhanced access to education, especially for girls in rural areas [63–65].

The finding of this study, which show that education played a key role in contributing to the observed change in contraceptive use in Zambia resonates with what has been reported in prior studies conducted in Rwanda, Ethiopia, Kenya and Cameroon [12, 17, 60, 61, 66]. This observation potentially suggests that educated women are more likely to make informed decisions on their reproductive health by adopting contraceptive use. Furthermore, their informed decisions may be because of the known maternal health benefits which accrue from utilising contraceptive methods [45, 46, 67]. On the other hand, the finding of this study on the role of education on contraceptive use change disagrees with the results of a study conducted by Yussuf and others in 2020, which reported no effect of education in driving contraceptive use change in Tanzania [62]. This finding from this current study showed that contraceptive behaviour change among women is facilitated by education in a significant way.

Other compositional changes that also made contribution to the observed increase in contraceptive use increase were the reduction in proportion of women who desired to have 6 or more children and the reduction in the proportion of women who experienced child mortality. The observation that a decline in the percentage of women who experience child mortality is connected with increased contraceptive use lends credence to the idea that there is a replacement effect in the correlation between fertility and child survival. In other words, experience of death of a child can lead to woman abandoning contraceptive use in order to replace a dead child [68–70]. This finding is consistent with what was found by studies conducted in Kenya and Ethiopia which reported that reduced experience of child death among women had a compositional effect on changes in contraceptive use [60, 66]. The finding of this study implies that improving child health strategies further, will help to accelerate on contraception uptake in the country.

Contrary to the expectation, the increase in proportion of women accessing family planning messages through media sources had no significant contribution effect to the contraceptive use increase in Zambia. This is despite many studies in SSA in showing evidence that women who accessed family planning message through mass-media family had increased likelihood of using contraceptive methods [71–76]. A recent study in Rwanda also reported similar findings [12]. On the other hand, Yussuf and others (2020) in Tanzania found that an increase in the proportion of women who heard family planning messages via media channels had a

significant positive contribution to contraceptive use change [62]. The finding of this study could suggest the need to redesign media family planning messages to make them more effective.

The findings of this study have implications for consideration to redesign the family planning communication campaigns in order to make them more effective. This is because health promotion campaigns have in other settings shown potential to enhance the self-esteem and authority of women to seek maternal health services. Furthermore, strategies that have proved to create an impact in influencing contraceptive use change, need enhancement so they can create more impact.

## Study limitations

The study had a number of limitations, first, since women in DHSs are usually asked to self-report information about contraceptive use, therefore, there is a possibility of under or over reporting of the event which could be subject to social desirability bias or lead to measurement error. Second, women in the DHSs report events that happened in the past, there is therefore a possibility of recall bias. Third, the analysis did not include some variables, such as household wealth status, visiting the health facility in the last 12 months, told about FP at the health facility and experienced pregnancy loss. Although these variables were reported in other studies to have influenced contraceptive use change, they were not analysed in this study because they were not collected in the 1992 DHS, which was the baseline for this analysis. Despite the limitations, this is the first study in Zambia to examine contraceptive use transition in Zambia by applying decomposition analysis approach. Further, the study utilised nationally representative data from the six cross-sectional datasets collected using the Zambia Demographic and Health Survey. Thus, findings of this current study can be generalised to the entire population of sexually active women aged 15–49 years in the country.

## Conclusion

The study has established that changes in women's contraceptive behaviour are responsible for the large share of the ongoing steady rise in the usage of contraceptives. Our study findings may reflect the need for further strengthening of FP interventions that are aimed at enhancing women's acceptance and utilisation of contraception services and commodities. The study further suggests the need for increased targeted evidence-based interventions such as sustained investment for the education sector and child health interventions in order to continue the rising trends in the use of contraceptives methods. Additionally, there is need to enhance the implementation of community-based maternal education and outreach programs targeting less educated women. Such an intervention aims to provide targeted information about contraception, family planning, and reproductive health at community level. Furthermore, there is a need for policy makers and programme implementers to consider designing interventions with an emphasis on promoting desire for a small family size to further increase uptake of contraception in Zambia.

## Acknowledgments

We appreciate the Zambia Statistics Agency, Ministry of Health, ICF and other partners involved in Zambia DHS program.

## Author Contributions

**Conceptualization:** Million Phiri, Clifford Odimegwu.

**Data curation:** Million Phiri, Yemi Adewoyin.

**Formal analysis:** Million Phiri, Yemi Adewoyin.

**Methodology:** Million Phiri.

**Project administration:** Million Phiri.

**Software:** Million Phiri.

**Supervision:** Clifford Odimegwu.

**Validation:** Clifford Odimegwu.

**Writing – original draft:** Million Phiri, Yemi Adewoyin.

**Writing – review & editing:** Million Phiri, Clifford Odimegwu, Yemi Adewoyin.

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
