## [Decision Letter · Decision Letter 0]

12 Dec 2023

PONE-D-23-29924A decomposition analysis of context of contraceptive use transition in Zambia (1992-2018)PLOS ONE

Dear Dr. Phiri,

Thank you for submitting your manuscript to PLOS ONE. After careful consideration, we feel that it has merit but does not fully meet PLOS ONE’s publication criteria as it currently stands. Therefore, we invite you to submit a revised version of the manuscript that addresses the points raised during the review process. Two expert reviewers provide their suggestions for revision. Regarding comment of reviewer 2 on the lack of changes in overall CP in the latest surveys, you document that while overall CP does not change much, there is a sharp change in terms of methods used. It would strenghthen the paper if you could link together your description of increased investment in SRH, the changing composition of methods, probably connected to contraceptive availability and hypothesize and/or interpret how this changing supply could affect the differential use according to sociodemographic variables that you are studying. Please submit your revised manuscript by Jan 26 2024 11:59PM. If you will need more time than this to complete your revisions, please reply to this message or contact the journal office at plosone@plos.org. Please include the following items when submitting your revised manuscript:A rebuttal letter that responds to each point raised by the academic editor and reviewer(s). You should upload this letter as a separate file labeled 'Response to Reviewers'.A marked-up copy of your manuscript that highlights changes made to the original version. You should upload this as a separate file labeled 'Revised Manuscript with Track Changes'.An unmarked version of your revised paper without tracked changes. You should upload this as a separate file labeled 'Manuscript'.

We look forward to receiving your revised manuscript.

Kind regards,

José Antonio Ortega, Ph.D.

Academic Editor

PLOS ONE

Journal Requirements:

Reviewers' comments:

Reviewer's Responses to Questions

**Comments to the Author**

1. Is the manuscript technically sound, and do the data support the conclusions?

Reviewer #1: Yes

Reviewer #2: Yes

2. Has the statistical analysis been performed appropriately and rigorously? 

Reviewer #1: Yes

Reviewer #2: Yes

3. Have the authors made all data underlying the findings in their manuscript fully available?

Reviewer #1: Yes

Reviewer #2: Yes

4. Is the manuscript presented in an intelligible fashion and written in standard English?

Reviewer #1: Yes

Reviewer #2: Yes

5. Review Comments to the Author

Reviewer #1: The manuscript's structure is meticulously organized, allowing for a seamless flow of ideas and a coherent presentation of the research findings. However, I want to redirect the authors to the following questions and comments.

Questions

1.) As you stated in the manuscript (Figure 3), much of the significant changes in contraceptive use occurred in the period 1992-2007, that is, a 24% increase has been recorded. Ever since 2007, the change has been sluggish and not worthy of decomposition. In my opinion, the reasons for the stall or the slower progress of contraceptive use in the recent decade (2007-2018) should be excavated and your style of analysis will neglect this fact when you emphasize only on the general rise. Why won't you focus on the decomposition of the abovementioned period 1992 to 2007? Otherwise, there is no point of using all the six DHS data for the decomposition. The first and the last would suffice.

2.) As you have indicated in the manuscript, the population of interest in your analysis are sexually active women. However, in the decomposition analysis, you used partner characteristics. This will restrict the analysis to married women or women currently in union. Are these two the same groups of women in Zambia? If not, please make sure that your analysis considered all sexually active women. Otherwise, you need to modify the title to fit to this consideration.

3.) In terms of relevance to family planning programs, traditional methods of contraception are of negligible importance. In the analysis, you considered folkloric and traditional methods in the same category as modern methods (see the dependent variable definition). Are these relevant in family planning programs in Zambia?

4.) In the method section of the manuscript, you indicated that the women data were used for analysis. Yet, community level variables were also considered in the analysis. How were these community variables constructed or computed unless you used the household data? For instance, for community education, you need to produce the data from household data. Similarly, you need the household data for community wealth/affluence. Please provide a statement to clarify how you constructed these measurements in the method section.

5.) How are the coefficient and endowment effects interpreted? You need to give your reader how the results are to be interpreted.

6.) In the conclusion part, you made the following statements:

'Changes in women’s contraceptive behaviour as a result of the continued FP promotion initiatives implemented over the 26-year period in Zambia, are responsible for the large share of the ongoing steady rise in the usage of contraceptives. Our study findings may reflect the need for further strengthening and up-scaling of family planning programmes in order to improve contraception services delivery, access and acceptance in Zambia in the quest to reduce high fertility and teenage

pregnancy in the country.'

You have not included program related variables in the decomposition analysis. What finding led you to make this conclusion?

7.) As a final note, the authors concluded that 'The study further suggests the need for increased targeted evidence-based

interventions such as sustained investment for the education sector ...'. This recommendation is okay for future generation of girls/women joining the reproductive age. However, the finding also suggests that being uneducated serves as a barrier to contraceptive use. Thus, 'what might we do to remedy the challenges of the current generation of women that are less educated?' is also a pointer to potential recommendation for programs and policy.

Comments

1.) Providing a theoretical underpinning for research will help to inform your research methodology, including the variable selection and interpretation of your findings. By so doing, you can relate your results back to the existing theories and concepts, either by confirming or challenging them, and provide a deeper understanding of the phenomenon under investigation.

2.) Please go through the document and refine the language. For instance, on the method section of the abstract, please see below

'Adjusted coefficients with their 95% confidence intervals and percentages of were used to present results'.

In addition, in the Introduction of the manuscript

'Change in utilisation of contraceptive methods can could result from improvement in women’s social context such as improvement in education and women empowerment or contraceptive behaviour change.'

3.) Please indicate the reference period for the decomposition analysis so that an appropriate interpretation can be made by your reader.

Reviewer #2: Comments to the authors

General comment: This study's writing and analysis are fine. But the tile is clumsy and needs revision.

Comment1. Based on the manuscript body, I would suggest rewriting the title as f:

1. Trend and socio-demographic factors in contraceptive use change among sexually active women in Zambia (1992-2018). A decomposition analysis

OR

2. Social context in contraceptive use change among sexually active women in Zambia (1992-2018). A decomposition analysis

Comment 2: In the abstract section, the authors stated that: changes in women’s compositional characteristics such as education, fertility preference, number of living children, and experience of child mortality were associated with the increase in contraceptive use prevalence. This result should be supported by the valid statistic test formed to arrive at this result (Adjusted Odds ratio or p-value).”

Comment 3: Discussion: I suggest adding one paragraph in the discussion section emphasizing the change in modern contraceptive use. That will contribute to the program improvement fusing on the modern contraceptive uptake in Zambia.

Comment 4. Some references are not well presented and should be written as follows:

25. United Nations Zambia. United Nations Sustainable Development

Cooperation Framework for the Republic of Zambia 2023 - 2027

[Internet]. Lusaka, Zambia; 2022 [cited 2023 Jul 20]. Available from: https://unsdg.un.org/

45. Ministry of Health. Reproductive Health Policy. Lusaka, Zambia: Ministry of Health; 2000 Jun.

Comment 5: The authors should add the link to access on the reference 45.

53. Hapompwe C. Sustainability of Universal Free Primary Education

Policy in the Face of Declining Education Financing in Lusaka,

Zambia. Afr J Educ Pract [Internet]. 2020 Jan 1 [cited 2023 Jun 29]; Available from:

https://www.academia.edu/43629975/Sustainability_of_Universal_Free_Primary_Education_Policy_in_the_Face_of_Declining_Education_Financing_In_Lusaka_Zambia

54. Masaiti G. Education in Zambia at Fifty Years of Independence and

Beyond: History, Current Status, and Contemporary Issues. 2019.

Conclusion: Well written with implications for program improvement of contraceptive use insight into the context in Zambia.

6. PLOS authors have the option to publish the peer review history of their article (what does this mean?). If published, this will include your full peer review and any attached files.

Reviewer #1: No

Reviewer #2: No

---

## [Author Response · Author response to Decision Letter 0]

24 Jan 2024

A detailed Rebuttal letter has been attached.

---

## [Decision Letter · Decision Letter 1]

29 Feb 2024

Social context in contraceptive use transition among sexually active women in Zambia (1992-2018): a decomposition analysis

PONE-D-23-29924R1

Dear Dr. Phiri,

We’re pleased to inform you that your manuscript has been judged scientifically suitable for publication and will be formally accepted for publication once it meets all outstanding technical requirements.

Kind regards,

José Antonio Ortega, Ph.D.

Academic Editor

PLOS ONE

Additional Editor Comments (optional):

The changes have addressed the issues raised and the two reviewers recommend acceptance.

Reviewers' comments:

Reviewer's Responses to Questions

**Comments to the Author**

1. If the authors have adequately addressed your comments raised in a previous round of review and you feel that this manuscript is now acceptable for publication, you may indicate that here to bypass the “Comments to the Author” section, enter your conflict of interest statement in the “Confidential to Editor” section, and submit your "Accept" recommendation.

Reviewer #1: (No Response)

Reviewer #2: All comments have been addressed

2. Is the manuscript technically sound, and do the data support the conclusions?

Reviewer #1: Yes

Reviewer #2: Yes

3. Has the statistical analysis been performed appropriately and rigorously? 

Reviewer #1: Yes

Reviewer #2: Yes

4. Have the authors made all data underlying the findings in their manuscript fully available?

Reviewer #1: Yes

Reviewer #2: (No Response)

5. Is the manuscript presented in an intelligible fashion and written in standard English?

Reviewer #1: Yes

Reviewer #2: Yes

6. Review Comments to the Author

Reviewer #1: (No Response)

Reviewer #2: (No Response)

7. PLOS authors have the option to publish the peer review history of their article (what does this mean?). If published, this will include your full peer review and any attached files.

Reviewer #1: **Yes: **Tariku Dejene

Reviewer #2: **Yes: **SIDIKIBA SIDIBE

---

## [Editor Report · Acceptance letter]

4 Apr 2024

PONE-D-23-29924R1 

PLOS ONE

Dear Dr. Phiri, 

I'm pleased to inform you that your manuscript has been deemed suitable for publication in PLOS ONE. Congratulations! Your manuscript is now being handed over to our production team.

Kind regards, 

on behalf of

Dr. José Antonio Ortega 

Academic Editor

PLOS ONE